# Effects of Psychotic Symptoms and Social Cognition on Job Retention in Patients with Schizophrenia in Korea

**DOI:** 10.3390/ijerph17082628

**Published:** 2020-04-11

**Authors:** Mihwa Han, Seong Sook Jun

**Affiliations:** 1Department of Nursing Science, Sunlin University, 30, 36beon-gil, Chogok-gil, Heunghae-eup, Pohang-si, Gyeongbuk 37560, Korea; mihwahanrn@naver.com; 2College of Nursing, Pusan National University, Beomeo-ri, Mulgeum-eup, Yangsan-si Gyeongnam, Busan 50612, Korea

**Keywords:** schizophrenia, social cognition, job retention, theory of mind, attribution style

## Abstract

This research examined the relationship between psychotic symptoms, social cognition, and job retention among people with schizophrenia in Korea. Participants (158 people with schizophrenia from 15 mental health institutions) were divided into two groups: those with a job retention period of less than six months (*n* = 75), and those with a job retention period of six months or more (*n* = 83). Participants completed a survey packet containing the Brief Psychiatric Rating Scale (BPRS), Global Assessment of Function (GAF) Scale, Interpersonal Relationship Functioning Assessment Scale, Basic Empathy Scale, Hinting Task, and Ambiguous Intention Hostility Questionnaire (AIHQ), and provided their job retention status. We used binomial logistic regression analysis to examine whether job retention was affected by participants’ demographic, clinical, and vocational characteristics, as well as the three components of social cognition, i.e., theory of mind, empathy, and attribution style. Results showed that theory of mind (ToM), attribution style, and psychotic symptoms explained 52.7% of the variance in job retention. A higher theory of mind means a higher ability to grasp the intentions of others. The higher theory of mind, the lesser attribution style, and the lesser psychotic symptoms were related to a longer period of job retention.

## 1. Introduction

Social cognition and job retention were found to be affected by the symptoms and functional restrictions of schizophrenia [1]. Social cognition refers to the mental operations underlying social interactions, which include the human ability to perceive the intentions and dispositions of others [2]. Social functioning deficits, which are common in people with schizophrenia, negatively affect many aspects of everyday life, including social relationships, occupational achievements, and independent living. Furthermore, Berger et al. [3] found that people with schizophrenia have poor social cognitive abilities, preventing them from perceiving the emotions and understanding the mental states of others.

These poor social cognitive abilities cause people with schizophrenia to be cautious about and suspicious of others, leading to problems in interpersonal relations and distorted situation awareness, rendering them unable to solve social problems, and having negative effects on job retention [4,5]. In line with this, Marwaha and Johnson [6] found that the employment rate among people with schizophrenia was the lowest across all assessed mental disorders, at 8%–20%, and that the chronic nature of the disorder means that occupational capabilities were markedly low.

To integrate into the local community and obtain economic and social independence, people must retain employment [7]. Novick et al. [8] carried out a 36-month follow-up study of 6643 outpatients with schizophrenia, and results from a comparative analysis of clinical and socio-demographic characteristics showed that 263 of them adjusted to local communities without recurrence but 6379 did not. Furthermore, the quality of life among people with schizophrenia who had, compared to those who did not have, positive professional experiences was relatively higher, and they were twice as likely to recover. Laurie and Brassington [9] stated that job retention is the ability of a person to work for a certain period in their local community, integrating and adjusting to society, and that, as the last step toward achieving the goal of independent living, it is a measure of assessing the success of occupational rehabilitation.

Tsang et al. [10] analyzed 62 research papers and stated that factors that influence the professional outcomes of people with schizophrenia include age, level of education, psychotic symptoms, general function, interpersonal functioning, and job history (whether or not they achieved occupational accomplishments). For demographic characteristics, those aged 25–45 years and who have higher levels of education are more likely to be hired and to retain their jobs [11]. The clinical characteristic of psychotic symptoms was reported as the cause of 56.7% of employment terminations [12]. Overall functioning and interpersonal functioning, that is, the functional status of people with schizophrenia, affect the ability to retain jobs and utilize diverse social support systems. People with schizophrenia who have long job retention periods and are successfully hired when faced with employment competition have high global and interpersonal functioning. In terms of occupation, premorbid job history was found to be related to the age of onset of schizophrenia, and faster recovery of function with good premorbid functioning, job experience, and later age of onset [13,14].

Harvey et al. [15] suggested that social cognition impacts the functional outcomes of people with schizophrenia. Social cognition has a positive correlation with factors such as occupation, residence, and marital status, as well as functional outcome, among people with schizophrenia [16]. However, despite numerous suggestions that social cognition and functional outcome may be correlated, there are few existing studies of this relationship.

To extend previous studies, we aimed to understand how job retention among people with schizophrenia is affected by the demographic characteristics of age, level of education, and age of onset; the clinical characteristics of psychotic symptoms, general functioning, and interpersonal relationship functioning; the vocational characteristics of premorbid job history, income, and working hours; and social cognition (i.e., the domains of social cognition include empathy, theory of mind, and attribution style). Empathy is defined as the ability to understand, as well as sympathize with, the experience and emotion of others [17]. The theory of mind refers to the ability of the mind to represent the mental states of others’ intentions. Attributional style refers to assigning causality to positive and negative events that may emphasize ambiguous situations [18]).

We examined the relationship between psychotic symptoms, social cognition, and job retention among people with schizophrenia in order to suggest a basis for establishing an intervention plan for stable job retention. Our specific purposes were as follows: (1) to obtain the demographic, clinical, and occupational characteristics of people with schizophrenia; (2) to examine the social cognition of people with schizophrenia; (3) to determine which of these factors affect the job retention of people with schizophrenia.

## 2. Material and Methods

### 2.1. Subjects and Procedure

Participants were 158 people with schizophrenia, who were aged between 19 and 65 years and registered in local community mental health institutions. All participants consented in writing to take part in this study, were diagnosed with schizophrenia according to psychiatric diagnosis criteria [19], and were taking antipsychotic drugs. They were all currently employed or previously worked for one day or more within the past two years. In order to obtain a homogeneous sample, people who were unable to read and answer the questionnaire, had language disorders, substance abuse problems, or intellectual disabilities, or who were diagnosed with either neurocognitive disorders or epilepsy were excluded.

Using the G * Power 3.1 program.(www.gpower.hhu.de) , the significance level (α) of 0.05 and the statistical power of 0.80 were input to calculate the number of subjects required for logistic regression analysis, which was found to be 153 people. In a previous study [20] on Korean mentally ill patients, 30% of respondents were poor or non-responsive. In this study, considering the 30% dropout rate of the previous study, questionnaires were distributed to 210 people, and 165 surveys were collected.

This research received approval from the local research ethics committee (IRB No. 05-2013-039), and we administered a preliminary survey using 11 people with schizophrenia from one mental health hospital rehabilitation center, two mental health promotion centers, and one psychosocial rehabilitation center. Afterward, the survey was revised and corrected to create the final version. Cronbach’s alpha internal consistency for the preliminary survey was 0.89.

We distributed 210 self-report surveys to the study participants, 51 of which were collected from three mental health hospital rehabilitation centers, 44 from seven mental health promotion centers, and 70 from five psychosocial rehabilitation centers; thus, 165 surveys (response rate = 78.5%) were collected from 15 institutions. After seven with missing responses were removed, 158 surveys were analyzed.

Thirty data collectors assisted with the study (five men, 25 women), including 22 mental health specialists, one nurse, and seven social workers. They had an average of five years and six months of experience in mental healthcare, and they were given training to ensure that they would not have difficulty in using the research tools employed in this study.

### 2.2. Measures

The demographic characteristics of this study were identified by six questions asking about the gender, age, onset of age, number of psychotic episodes, marital status, and level of education. Vocational characteristics were captured using four items: previous employment, occupation and employment type, income, and length of work day.

The clinical characteristics measured in this study, which comprised psychotic symptoms, general functioning, and interpersonal functioning, were obtained through blind collection by researchers in each institution.

Psychotic symptoms were measured with the 18-item Brief Psychiatric Rating Scale (BPRS), which is used to assess the severity of mental disorders [21]. Each item is measured on a five-point scale (0 = none to 4 = very severe), with lower scores meaning a lower severity of the symptom. Rubin and Overall [22] found in their study that, when the BPRS evaluator was a nurse, Cronbach’s alpha reliability coefficient was 0.97. The mental health professional data collectors conducted this survey via interviews and observation. Evaluation involved adding up the scores for each item, and the Cronbach’s alpha reliability in the study was 0.90.

The general functioning of people with schizophrenia was measured in this study with the Global Assessment of Function Scale [23]. It is widely used to assess the multi-axial system of Axis V mental disorders, and it was adopted to measure the psychopathology and functioning of people with schizophrenia [24]. It comprises one item for an overall assessment of an individual’s functioning, regardless of diagnosis. The evaluator subjectively assesses the participant’s symptoms and functioning on a scale from 0–100. A lower score denotes a lower global functioning of the participant [25]. Cronbach’s alpha internal consistency in this study was 0.81.

To assess the interpersonal functioning of people with schizophrenia, we used the 23-item Interpersonal Relationship Functioning Assessment Scale, which was adapted and revised by Kim and Han [26] from the Independent Living Skills Survey [27]. The score range was from one (almost no interpersonal functioning) to five (very good interpersonal functioning), with total scores ranging from 23 to 115 points. Higher scores indicate higher levels of interpersonal functioning. In Kim and Han’s study [26], Cronbach’s α reliability was 0.96, and, in this study, it was 0.93.

In this study, we measured three components of social cognition: empathy, theory of mind, and attribution style.

In order to measure empathy, this study used the 20-item Basic Empathy Scale, which is a self-report tool that was originally developed by Jolliffe and Farrington [28], and later translated into Korean, revised, and improved by Kang and Lee [29]. Responses are made on a five-point Likert scale (1 = strongly disagree to 5 = strongly agree), and the scale comprises nine items assessing the cognitive element (3, 6, 9, 10, 12, 14, 16, 19, 20) and 11 items assessing the emotional element (1, 2, 4, 5, 7, 8, 11, 13, 15, 17, 18). In Lee’s [30] study of people with schizophrenia, the Cronbach’s alpha reliability was 0.80. The Cronbach’s alpha reliability of the tool in this study was 0.71.

To measure theory of mind, this study used the Hinting Task, which was produced by Corcoran et al. [31]. It is a self-report survey that was devised to evaluate the ability to deduce hidden motives underneath indirect speech. This test comprises 10 vignettes, each describing a short conversation between two characters. Correct answers receive two points, correct answers with hints receive one point, and incorrect answers receive zero points, with the total score ranging from 0–20. In a study by Roberts [32], Cronbach’s alpha reliability was 0.65.

This study used an eight-vignette version of the tool that was adapted into Korean by Hur et al. [33], after excluding two stories that did not fit the cultural context. Total scores range from 0–16, and Cronbach’s alpha reliability in this study was 0.78.

To measure attribution style, we used the Korean Ambiguous Intention Hostility Questionnaire (K-AIHQ), which was adapted and revised by Chang et al. [34] based on the Ambiguous Intention Hostility Questionnaire [35], under the assumption that people with schizophrenia who exhibited paranoia would show biased social cognition. This is a self-report survey that evaluates hostile social cognitive bias by measuring hostility, blame (the tendency to hold others responsible), and aggression, based on 15 negative situations that lead to negative results. Participants read the situation and imagine it happening to them, before using a Likert scale (1 = not at all to 5 = very much) to rate the degree to which they believe that the other person acted with a motive toward the participant, and how much the participant blames the other. Evaluators assessed hostility and aggression through answers that were subjectively drawn up by the participants. Combs et al. [35] showed that the three items that measured blame had a Cronbach’s alpha internal consistency of 0.83–0.86, and Elnakeeb et al. [36] recorded a Cronbach’s α of 0.81–0.83. In a study of the Korean version of the scale, which was conducted by Chang et al. [34], Cronbach’s alpha of internal consistency was 0.53, 0.61, and 0.75 for the three subscales, whereas, in this study, it was 0.72, 0.77, and 0.78, respectively.

Cheadle and Morgan [37] defined successful professional rehabilitation as job retention of six months or more without a break of two weeks or more, regardless of whether the same job was held for the full time. In this study, the dependent variable of job retention was measured using this criterion, through distinguishing between those that held a job for six months or more and those that did not.

### 2.3. Statistical Analysis

Data analyses were carried out using SPSS (Windows, IBM: New York, NY, USA) Version 19.0. Through frequency analysis of the collected data, errors and normality of distribution were identified, and the reliability of the research tools was calculated. The participants were divided into groups of those with a job retention period of less than six months and those with job retention period of six months or more, and participants’ demographic and occupational characteristics were calculated via frequency and percentage. Differences in job retention periods according to demographic, clinical, and professional characteristics, as well as social cognition, were found through calculating the mean, standard deviation, chi-square (χ^2^), and *t*-test values. We carried out Pearson correlation analysis to assess correlations between factors. Variables that showed significant differences between the two groups were assessed with logistic regression analysis, to determine which factors impacted the participants’ job retention.

## 3. Results

### 3.1. Demographic Characteristics

As can be seen in Table 1, the differences in gender, age, age of onset of schizophrenia, the number of psychotic episodes, marital status, and level of education between the two job retention groups were not significant. In both groups, there were more men, and most were aged 30–40 years, developed schizophrenia in their 20s, had a single marital status, and obtained a high-school diploma. The number of psychotic episodes most often experienced was 3–6 episodes in both groups.

### 3.2. Clinical Characteristics

Table 2 shows the results of analysis of participants’ clinical characteristics. Members of the group with a job retention period under six months had more psychotic symptoms than those who held a job for six months or more, and they had lower global and interpersonal functioning. The differences between the two groups in terms of psychotic symptoms (*p* < 0.001), global functioning (*p* < 0.001), and interpersonal functioning (*p* = 0.007) were significant.

### 3.3. Vocational Characteristics 

As can be seen in Table 3, among the group that retained a job for six months or more, the two most common employment types were supported and independent employment, and occupation types were simple labor worker and production worker. This group also had a higher income and longer work hours in comparison to the group with a job retention period of less than six months. Differences between the two groups in terms of employment type (*p* = 0.011), income (*p* = 0.003), and work hours (*p* = 0.009) were significant.

### 3.4. Social Cognition

As can be seen in Table 4, those with a job retention period of six months or less had higher hostility, blame, and aggression scores, whereas those who had held a job for six months or more had higher theory of mind scores. In terms of social cognition, differences between the two groups in terms of theory of mind (*p* < 0.001), blame (*p* = 0.001), and hostility (*p* = 0.005) were significant.

### 3.5. Results of Binary Logistic Regression Analyses

To establish which factors affect job retention among people with schizophrenia, all of the above variables that showed significant differences between job retention groups were employed as independent variables in logistic regression analyses. The results are shown in Table 5.

The Hosmer–Lemeshow test, which assesses the goodness of fit for logistic regression analyses, showed a *p*-value of 0.539, which demonstrates that the model developed (χ² = 79.434, *p* < 0.001) fits the regression model data well. The explanatory power of the independent variables in the regression model was 52.7%, and the classification accuracy of the two groups in the regression model was 82.9%. 

Among the nine independent variables employed in the regression model, type of employment was set as a dummy variable. In terms of the impact of the five types of employment on job retention, sheltered employment had a negative effect. Sheltered employment was then categorized and employed as a type of employment in the regression model, but examination of its impact along with other factors revealed that it was not significant.

Furthermore, theory of mind (*p* < 0.001), hostile attribution style (*p* = 0.023), and psychotic symptoms (*p* = 0.047) significantly affected job retention. For the group with a job retention period of six months or more, the Wald statistic was 28.090 and the significance probability was 0.000, with Exp(B)(odds ratio) = 1.468, showing that, when the score for theory of mind under social cognition increased by one, there was a 1.468-fold increase in job retention of six months or more. When the score for psychotic symptoms increased by one, and hostility under attribution style increased by one, there was a 0.948-fold and a 0.399-fold decrease in job retention of six months or more.

## 4. Discussion

This study examined the factors that impact job retention among people with schizophrenia, to suggest a basis for establishing intervention plans for stable job retention. We observed significant differences between participants who did and did not hold a job for six months or more, in terms of the clinical characteristics of psychotic symptoms, global functioning, and interpersonal functioning; the vocational characteristics of type of employment, income, and work hours; theory of mind as related to social cognition; and hostility and blame attribution perceptions. These nine variables explained 52.7% of the variance in job retention, with theory of mind having the biggest impact, and then hostile attribution style and psychotic symptoms.

Regarding demographic characteristics, most participants were in their 30s and 40s, with no differences between the two groups. Hofer et al. [38] stated that 25–45 years of age is when vocational functioning is highest and when there is good maintenance of cooperative relationships with co-workers, and Burke-Miller et al. [39] stated that this age range is when it is best to achieve vocational success, further commenting that people with schizophrenia in this age group have more of an advantage in competitive employment situations compared to those aged 45 years or above. The results of this study support these findings regarding those in the age group of 30–40 years. Upon reaching the age of 45 years, the probability of being hired in competitive employment situations decreases, and there are more instances of job retention in simple and unskilled professions [12]. This is related to the argument that those aged 45 years or above have lower cognitive function, intellectual ability, and sense of judgment compared to younger people [40].

Previous studies stated that lower levels of education lead to lower vocational achievement [41,42], but there was no significant difference observed in relation to level of education in this study. This is related to the employment environment surrounding people with schizophrenia in Korea, particularly the difficulty that people with schizophrenia experience when trying to return to their former jobs after the onset of the disorder [43]. Most people with schizophrenia have unskilled jobs, such as laboring, manufacturing, or sales/service. Although this may be due to social bias, it also appears to be a result of the disorder becoming chronic, in addition to schizophrenia’s characteristic symptoms [44].

We found no significant difference in the age of onset of schizophrenia between the two job retention groups. One previous study [10] showed that a lower age of onset of schizophrenia negatively affects job retention because of reduced opportunities to acquire social skills and participate in a normal curriculum. This implies that there was no significant difference in the age of onset between the occupational maintenance group and the non-maintenance group among the study subjects. This is because the degree of social skill acquisition and education were related to vocational proficiency. This led to the similar results seen in the two groups representing the occupation type.

In terms of clinical characteristics, psychotic symptoms had a significant effect on job retention period. The BPRS, which was used in this study to measure psychotic symptoms, was administered in a previous study by Lachar et al. [45] of 1415 people with schizophrenia, who showed an average score of 24.3 ± 13.2 during the acute phase, and 12.1 ± 13.4 during the recovery phase. In this study, our six months or more job retention group scored 12.5 ± 8.5, which is closer to the recovery phase scores of Lachar et al. [45] than the less than six months job retention group, which scored 19.9 ± 11.8. Srinivasan and Tirupati [46] stated that psychotic symptoms affect the vocational outcomes of people with schizophrenia, which is supported by the present study results. In a five-year longitudinal study, Russinova et al. [12] found that 56.7% of people with schizophrenia quit their jobs due to worsening of psychotic symptoms and concluded that continuous symptom-related intervention is needed.

We measured global functioning with a scale that assesses mental health symptoms and social/professional functioning in people with schizophrenia, and we found that higher scores were associated with longer job retention. The global functioning score was significantly different for the groups with less than or more than six months of job retention, at 67.0 ± 9.3 and 58.0 ± 13.5 points, respectively, with lower scores indicating worse functioning. Higher global functioning reflects a superior ability to acquire social skills, and reduced cognitive damage in people with schizophrenia, both of which have positive effects on job retention [47].

Interpersonal functioning, which enables socialization with others, thereby allowing a person to call for help when needed and proactively participate in social events, making it necessary in social situations [10], also differed between the two groups. Kim et al. [48] measured interpersonal functioning in people with schizophrenia from a local community mental health institution and reported an average score of 78.22 ± 9.31, compared to those whose schizophrenic symptoms were in the remission phase, who were found in a later study to have an average score of 76.40 ± 9.21 [48]. In this study, the interpersonal functioning score for the six months or more job retention group was 80.5 ± 14.3, and that for the less than six months job retention group was 74.2 ± 14.5; thus, the interpersonal functioning of the former group exceeded the average of people with schizophrenia whose symptom status was in the remission phase.

In terms of vocational characteristics, the groups showed significant differences in terms of types of employment, income, and work hours per day. A longer job retention period was associated with a higher rate of independent employment and a lower rate of sheltered employment, which aligns with previous study findings that higher income and longer work hours were associated with longer job retention [49,50]. Marwaha and Johnson [6] reported that, as psychotic symptoms relapsed and intensified, occupational skills became significantly reduced and led to job discontinuation. Moreover, from this study, it is evident that the psychotic symptoms of the subjects who remained in the job for more than six months were better than those who did not.

In terms of social cognition, the two groups showed significant differences in theory of mind, hostility under attribution style, and blame, which aligns with the results of Penn et al. [51] and Horan et al. [16]. Furthermore, theory of mind under social cognition among people with schizophrenia involves the ability to understand others’ emotional and mental states. The hostile attribution style is related to judging the cause of motivation of others to be with oneself when attempting to guess the motivation of others. In this way, people recognize and understand external stimuli in social situations, and this also seems to be significantly related to job retention. Therefore, there is a need for further studies on the ability to understand the emotional states of others, and for interventions to prevent perceptions of hostility in relation to the motives of others.

Hur et al. [33] developed the Korean version of the Hinting Task, which measures theory of mind, and reported an average score for people with schizophrenia in the stable plateau phase of 9.52 ± 3.16, and for those without a history of mental health disorders of 14.43 ± 1.22. Our participants in the six months or more job retention group scored 12.1 ± 2.3, and those in the less than six months job retention group scored 7.9 ± 3.9, demonstrating that the former group had a higher theory of mind score than people with schizophrenia in the stable plateau phase from previous studies.

In the case of blame, as a sub-category of attribution style, Jeon et al. [52] assessed 263 non-disabled people with an average age of 21.1 years, and reported average scores for motive, anger, and blame of 1.56 ± 0.46, 2.43 ± 0.58, and 1.76 ± 0.28, respectively. Lee [30] assessed people with schizophrenia with an average age of 39.6 years, who participated in vocational rehabilitation through supported employment, and reported motive, anger, and blame scores of 2.53 ± 1.03, 4.15 ± 1.52, and 2.29 ± 0.78, respectively. In this study, the six months or more job retention group had scores of 1.48 ± 0.54, 2.38 ± 0.75, and 1.74 ± 0.45, respectively, and the less than six months job retention group scored 1.77 ± 0.72, 2.79 ± 0.80, and 1.88 ± 0.68, respectively. In this study, the job retention group over six months was found to have a lower attribution style score than the group with a job retention period of less than six months. However, both groups had scores similar to those of non-disabled people in previous studies and lower than those of people with supported employment.

However, empathy, as assessed with a self-report survey, in which the participant assesses their own degree of empathy, did not significantly differ between the two groups. People with schizophrenia show low emotional perspective scores in relation to understanding or attempting to understand another person’s point of view, but high levels of identification or empathizing with characters in films or novels [53]. This may be due to people with schizophrenia finding it difficult in the real world to consider the position of others and think of ways to change their perspectives, meaning that they have a greater tendency to retreat into imaginary worlds. In this study, the empathy score of the less than six months job retention group (59.0 ± 10.1) was slightly but not significantly higher than that of the six months or more job retention group (58.7 ± 5.4).

## 5. Conclusions

In conclusion, by analyzing the difference between groups who had more vs. less than six months of job retention, we showed that participants with longer job retention periods had fewer psychotic symptoms, and better global and interpersonal functioning. Their work hours per day were also longer, income was higher, rate of independent employment was higher, and rate of supported employment was lower. In terms of social cognition, the longer job retention group had better theory of mind abilities than the shorter job retention group, which enabled them to understand and deduce other people’s mental states, and lower hostility, indicating less suspicion of others in incidents in interpersonal relationships.

This study has importance in that we showed that social cognition should be included in job retention interventions in relation to the influence of types of employment, psychotic symptoms, and function. Specifically, for stable job retention among people with schizophrenia, an intervention program is needed to improve social cognitive abilities, particularly in relation to addressing psychotic symptoms and attribution bias under social cognition, and to improving flexibility of thought and empathy in interpersonal relationships.

A limitation in this study relates to our use of a sample of Korean people living in specific areas; thus, the study results may not be able to be generalized to other people living in different areas in Korea or other countries. In addition, it is necessary to identify physical illnesses, such as metabolic syndrome and family support, and suggest follow-up studies to determine the degree of impact of job retention on the quality of life of schizophrenic patients.

Future studies should examine the influence of changes in social cognition on job retention by measuring the participants’ social cognition, and by measuring the relationship between job retention and social cognition over time.

## Figures and Tables

**Table 1 ijerph-17-02628-t001:** Participants’ demographic characteristics.

Variables	Categories	Total (*n*)	Less Than 6 Months		More Than 6 Months		χ²	*p*
			*n* (%)	M ± SD	*n* (%)	M ± SD		
Gender	Male	106	49 (46.2)		57 (53.8)		0.199	0.655
	Female	52	26 (50.0)		26 (50.0)			
Age (years)	20–29	21	13 (61.9)		8 (38.1)		2.830	0.419
	30–39	54	27 (50.0)		27 (50.0)			
	40–49	50	21 (42.0)		29 (58.0)			
	50–65	33	14 (42.4)		19 (57.6)			
				39.5 ± 9.7		41.6 ± 9.0		
Age at onset of Schizophrenia (years)	5–19	36	19 (52.8)		17 (47.2)		3.005	0.391
	20–29	85	43 (50.6)		42 (49.4)			
	30–39	31	11 (35.5)		20 (64.5)			
	40–50	6	2 (33.3)		4 (66.7)			
				23.9 ± 6.8		25.8 ± 7.4		
Number of psychotic episodes *	0–2	58	25 (43.1)		33 (56.9)		6.331	0.275
	3–6	66	28 (42.4)		38 (57.6)			
	≥7	32	21 (65.6)		11 (34.4)			
Marital status	Single	129	62 (48.1)		67 (51.9)		6.583	0.160
	Married	12	5 (41.7)		7 (58.3)			
	Divorced	17	8 (47.1)		9 (52.9)			
Level of education (years)	9–11	17	11 (64.7)		6 (35.3)		4.067	0.254
	12–15	101	49 (48.5)		52 (51.5)			
	≥16	40	15 (37.5)		25 (62.5)			

Note: *N* = 158; * except the missing data that are not reported (*N* = 156).

**Table 2 ijerph-17-02628-t002:** Participants’ clinical characteristics.

Variables	Total (M ± SD)	Less Than 6 Months	More Than 6 Months	*t*	*p*
		M ± SD	M ± SD		
BPRS	16.1 ± 10.8	19.9 ± 11.8	12.5 ± 8.5	4.478	<0.001 ***
GAF	62.8 ± 12.3	58.0 ± 13.5	67.0 ± 9.3	−4.775	<0.001 ***
Interpersonal functioning	77.6 ± 14.7	74.2 ± 14.5	80.5 ± 14.3	−2.730	0.007 **

Note: *N* = 158. BPRS: Brief Psychiatric Rating Scale, GAF: Global Assessment of Functioning Scale; ** *p* < 0.01, *** *p* < 0.001.

**Table 3 ijerph-17-02628-t003:** Participants’ vocational characteristics.

Variables	Categories	Total (*n*)	Less Than 6 Months		More Than 6 Months		χ²/*t*	*p*
			*n* (%)	M ± SD	*n* (%)	M ± SD		
Previous employment	Yes	93	41 (44.1)		52 (55.9)		1.037	0.308
	No	65	34 (52.3)		31 (47.7)			
Occupation type	Professionals	1	0 (0)		1 (100.0)		4.367	0.359
	Administrative worker	7	2 (28.6)		5 (71.4)			
	Service sales worker	37	14 (37.8)		23 (62.2)			
	Production worker	53	27(50.9)		26(49.1)			
	Simple labor worker	60	32(53.3)		28(46.7)			
Employment type	Sheltered	33	23 (69.7)		10 (30.3)		14.815	0.011 *
	Public service	16	10 (62.5)		6 (37.5)			
	Supported	50	22 (44.0)		28 (56.0)			
	Transitional	27	12 (44.4)		15 (55.6)			
	Independent	32	8 (25.0)		24 (75.0)			
Income † (KRW)	≤20	61	38 (62.3)		23 (37.7)		18.044	0.003 **
	21–40	22	11 (50.0)		11 (50.0)			
	41–80	31	11 (35.5)		20 (64.5)			
	≥81	38	10 (26.3)		28 (73.7)			
			29.9 ± 36.4		55.9 ± 41.7			
Length of work day (hours)	1–2	24	19 (79.2)		5 (20.8)		11.653	0.009 **
	3–4	26	12 (46.2)		14 (53.8)			
	5–8	74	30 (40.5)		44 (59.5)			
	≥9	34	14 (41.2)		20 (58.8)			
			5.7 ± 3.1		7.1 ± 3.4			

Note: *N* = 158; † except the six people who are currently unemployed (*N* = 152); KRW: Korean Republic Won; * *p* < 0.05, ** *p* < 0.01.

**Table 4 ijerph-17-02628-t004:** Participants’ social cognition.

Variable	Total (M ± SD)	Less Than 6 Months	More Than 6 Months	*t*	*p*
		M ± SD	M ± SD		
BES	58.9 ± 7.9	59.0 ± 10.1	58.7 ± 5.4	0.210	0.834
Hinting task	10.1 ± 3.8	7.9 ± 3.9	12.1 ± 2.3	−7.993	<0.001 ***
AIHQ					
Hostility	1.62 ± 0.65	1.8 ± 0.72	1.5 ± 0.54	2.855	0.005 **
Blame	2.57 ± 0.80	2.8 ± 0.80	2.4 ± 0.75	3.264	0.001 **
Aggression	1.81 ± 0.57	1.9 ± 0.68	1.7 ± 0.45	1.557	0.122

Note: *N* = 158. BES: Basic Empathy Scale, AIHQ: Ambiguous Intentions Hostility Questionnaire; ** *p* < 0.01, *** *p* < 0.001.

**Table 5 ijerph-17-02628-t005:** Summary of logistic regression analyses for variables predicting job retention.

Predictor	B	SE	Wald	*p*	Exp(B)	95% CI
Clinical Characteristics						
BPRS	−0.053	0.027	3.935	0.047	0.948	0.900, 0.999
GAF	0.013	0.025	0.265	0.607	1.013	0.965, 1.063
Interpersonal functioning	−0.012	0.017	0.509	0.476	0.988	0.955, 1.022
Vocational Characteristics						
Employment type	−0.937	0.686	1.867	0.172	0.392	0.102, 1.502
Income	0.003	0.006	0.203	0.652	1.003	0.990, 1.015
Length of work day (hours)	−0.037	0.080	0.219	0.640	0.963	0.824, 1.126
Social Cognition						
Hinting task	0.384	0.072	28.090	<0.001	1.468	1.274, 1.692
AIHQ						
Hostility	−0.920	0.403	5.200	0.023	0.399	0.181, 0.879
Blame	−0.252	0.351	0.515	0.473	0.777	0.391, 1.547
Constant	−0.358	2.229	0.026	0.872	0.699	
Hosmer and Lemeshow: χ² = 6.976, df = 8, *p* = 0.539; −2log likelihood: 139.195 (*p* < 0.001)						
Model summary: χ² = 79.434 (df = 9/*p* < 0.001); Nagelkerke *R*² = 0.527; correct classification: 82.9%						

Note: *N* = 158; BPRS: Brief Psychiatric Rating Scale, GAF: Global Assessment of Function scale, AIHQ: Ambiguous Intentions Hostility Questionnaire, SE: standard error, CI: confidence interval.

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
