# Peer review of "Effects of Psychotic Symptoms and Social Cognition on Job Retention in Patients with Schizophrenia in Korea"

_ijerph, 2020, doi:10.3390/ijerph17082628_

Round 1

Reviewer 1 Report

The authors have done a great job assessing factors influencing job retention in patients with schizophrenia. Overall, the manuscript is well written. However, there are some concerns I would like to point out:

General Comments:

1) The authors report age of onset of schizophrenia and other interesting patient characteristics that are play a role in disease outcomes. A further important factor is number of psychotic episodes – has this been assessed in this study? I would expect this to affect job retention as well.

2) I think it would be interesting for the authors to present/mention briefly the clinical characteristics of the patients according to their employment type (see comment 6 below)

Specific Comments:

Page 2, line 70: The authors write: “social cognition (i.e. empathy—which is defined …”. This might be misinterpreted as: social cognition = empathy. Please clarify.

Page 5, Table 1: Authors should consider reporting percentages horizontally to increase readability of the table, e.g. Gender – Male  - 106  - 49(46%)  -75 (53%) , since the main comparison here is employment duration and not gender.

Page 6, Table 3:

1) Does “work history” mean previous employment? Please consider replacing.

2) employment type: is independent employment missing here?

3) Please consider reporting percentages horizontally here, as well

4) Page 9, lines 265-268: Please provide references. Especially in regard to the statement that patients with schizophrenia have unskilled jobs: is this the case also before the disease onset? Did this study collect data about employment type before disease onset?

5) Page 9, lines 274-275: “thus, it can be seen that rehabilitation had a more direct impact on job retention.” How can this be seen? Please clarify.

6) Page 10, lines 304-308: Probably because these patients had less symptoms? Please add a sentence or two on that.

Author Response

Comments and Suggestions for Authors

The authors have done a great job assessing factors influencing job retention in patients with schizophrenia. Overall, the manuscript is well written. However, there are some concerns I would like to point out:

General Comments:

1) The authors report age of onset of schizophrenia and other interesting patient characteristics that are play a role in disease outcomes. A further important factor is number of psychotic episodes – has this been assessed in this study? I would expect this to affect job retention as well.

Response

Thank you for pointing this out. I had omitted adding this to the table to make the presentation brief, but based on your comment, I realized that it is necessary to include the number of psychotic episodes, and I have added it to Table 1.

Page 4, line 192~Page 5, line 196

As can be seen in Table 1, the differences in gender, age, age of onset of schizophrenia, the number of psychotic episodes, marital status, and level of education between the two job retention groups were not significant. In both groups, there were more men, and most were aged 30–40 years, had developed schizophrenia in their 20s, their marital status was “single,” and they had obtained a high school diploma. The number of psychotic episodes most often experienced was 3-6 episodes in both groups.

Table 1.

2) I think it would be interesting for the authors to present/mention briefly the clinical characteristics of the patients according to their employment type (see comment 6 below)

Response

Thank you for this suggestion. The response to this can be seen in the response added to comment 6. which outlines the clinical characteristics of patients based on their employment type.

Page 10, line 311~318

In terms of vocational characteristics, the groups showed significant differences in terms of types of employment, income, and work hours per day. A longer job retention period was associated with a higher rate of independent employment and a lower rate of sheltered employment, which aligns with previous study findings that higher income and longer work hours were associated with longer job retention (Bond et al., 2008; Park & Park, 2019).

Marwaha and Johnson (2004) report that as psychotic symptoms relapsed and intensified, occupational skills became significantly reduced, and led to job discontinuation. Also, from this study, it is eviednt that the psychotic symptoms of the subjects who remained in the job for more than 6 months were better than those who did not.

Specific Comments:

Page 2, line 70: The authors write: “social cognition (i.e. empathy—which is defined …”. This might be misinterpreted as: social cognition = empathy. Please clarify.

Response

Thank you for pointing this out. The definition of social cognition has been added to page 1, line 27, and a description of the domains of social cognition has been added to page 2, line 70.

Page 1. line 29~31

Social cognition and job retention have been found to be affected by the symptoms and functional restrictions of schizophrenia (Silberstein & Harvey, 2019). Social cognition refers to the mental operations underlying social interactions, which include the human ability to perceive the intentions and dispositions of others (Brothers, 1990).

Page 2, line 74~78

To extend previous studies, we aimed to understand how job retention among people with schizophrenia is affected by the demographic characteristics of age, level of education, and age of onset; the clinical characteristics of psychotic symptoms, general functioning, and interpersonal relationship functioning; the vocational characteristics of premorbid job history, income, and working hours; and social cognition (i.e. The domains of social cognition include empathy, Theory of Mind, and attribution style. Empathy is defined as the ability to understand as well as sympathise with the experience and emotion of others (Van Donkersgoed et al., 2015). The theory of the mind refers to the ability of the mind to represent the mental states of others’ intentions. Attributional style refers to assigning causality to positive and negative events that may emphasize ambiguous situations (Green et al. 2008).

Page 5, Table 1: Authors should consider reporting percentages horizontally to increase readability of the table, e.g. Gender – Male  106  49(46%) Female-75 (53%) , since the main comparison here is employment duration and not gender.

Response

Thank you. It has been corrected.

Table 1.

Table 1. Participants’ Demographic Characteristics.

Variables

Categories

Total(n)

Less than 6 months

More than 6 months

χ²

p

n (%)

M ± SD

n (%)

M ± SD

Gender

Male

106

49 (46.2)

57 (53.8)

0.199

0.655

Female

52

26 (50.0)

26 (50.0)

Age (years)

20–29

21

13 (61.9)

8 (38.1)

2.830

0.419

30–39

54

27 (50.0)

27 (50.0)

40–49

50

21 (42.0)

29 (58.0)

50–65

33

14 (42.4)

19 (57.6)

39.5 ± 9.7

41.6 ± 9.0

Age at onset of

schizophrenia

(years)

5–19

36

19 (52.8)

17 (47.2)

3.005

0.391

20–29

85

43 (50.6)

42 (49.4)

30–39

31

11 (35.5)

20 (64.5)

40–50

6

2 (33.3)

4 (66.7)

23.9 ± 6.8

25.8 ± 7.4

Number of psychotic episodes*

0-2

58

25 (43.1)

33(56.9)

6.331

0.275

3-6

66

28(42.4)

38(57.6)

≥7

32

21(65.6)

11(34.4)

Marital status

Single

129

62 (48.1)

67 (51.9)

6.583

0.160

Married

12

5 (41.7)

7 (58.3)

Divorced

17

8 (47.1)

9 (52.9)

Level of education (years)

9–11

17

11 (64.7)

6 (35.3)

4.067

0.254

12–15

101

49 (48.5)

52 (51.5)

≥ 16

40

15 (37.5)

25 (62.5)

Note. N = 158. *except the missing data that are not reported (N=156).

Page 6, Table 3:

1) Does “work history” mean previous employment? Please consider replacing.

Response

Thank you. It has been corrected.

Table 3.

Table 3. Participants’ Vocational Characteristics

Variables

Categories

Total (n)

Less than 6 months

More than 6 months

χ² / t

p

n (%)

M ± SD

n (%)

M ± SD

Previous employment

Yes

93

41 (44.1)

52 (55.9)

1.037

0.308

No

65

34 (52.3)

31 (47.7)

2) employment type: is independent employment missing here?

Response

Thank you for pointing this out. Employment type and Occupation type have been added.

Page 6, line 207-208

As can be seen in Table 3, among the group that had retained a job for 6 months or more, the two most common employment types were supported and independent employment, and occupation types were simple labour worker and production worker.

Table 3.

Table 3. Participants’ Vocational Characteristics

Variables

Categories

Total (n)

Less than 6 months

More than 6 months

χ² / t

p

n (%)

M ± SD

n (%)

M ± SD

Occupation type

Professionals

1

0 (0)

1 (100.0)

4.367

0.359

administrative worker

7

2 (28.6)

5 (71.4)

service sales worker

37

14 (37.8)

23 (62.2)

production worker

53

27(50.9)

26(49.1)

simple labour worker

60

32(53.3)

28(46.7)

Employment type

Sheltered

33

23 (69.7)

10 (30.3)

14.815

0.011*

Public service

16

10 (62.5)

6 (37.5)

Supported

50

22 (44.0)

28 (56.0)

Transitional

27

12(44.4)

15(55.6)

Independent

32

8(25.0)

24(75.0)

3) Please consider reporting percentages horizontally here, as well

Response

Thank you. It has been added as per your suggestion.

Table 3.

Table 3. Participants’ Vocational Characteristics

Variables

Categories

Total (n)

Less than 6 months

More than 6 months

χ² / t

p

n (%)

M ± SD

n (%)

M ± SD

Previous employment

Yes

93

41 (44.1)

52 (55.9)

1.037

0.308

No

65

34 (52.3)

31 (47.7)

Occupation type

Professionals

1

0 (0)

1 (100.0)

4.367

0.359

administrative worker

7

2 (28.6)

5 (71.4)

service sales worker

37

14 (37.8)

23 (62.2)

production worker

53

27(50.9)

26(49.1)

simple labour worker

60

32(53.3)

28(46.7)

Employment type

Sheltered

33

23 (69.7)

10 (30.3)

14.815

0.011*

Public service

16

10 (62.5)

6 (37.5)

Supported

50

22 (44.0)

28 (56.0)

Transitional

27

12(44.4)

15(55.6)

Independent

32

8(25.0)

24(75.0)

Income* (KRW)

≤20

61

38 (62.3)

23 (37.7)

18.044

0.003**

21–40

22

11 (50.0)

11 (50.0)

41–80

31

11 (35.5)

20 (64.5)

≥81

38

10 (26.3)

28 (73.7)

29.9 ± 36.4

55.9 ± 41.7

Length of work day (hours)

1–2

24

19 (79.2)

5 (20.8)

11.653

0.009**

3–4

26

12 (46.2)

14 (53.8)

5–8

74

30 (40.5)

44 (59.5)

≥9

34

14 (41.2)

20 (58.8)

5.7 ± 3.1

7.1 ± 3.4

Note. N = 158. *except the six people who are currently unemployed (N = 152). *p<0 .05, ** p<0 .01, *** p<0 .001

4) Page 9, lines 265-268:

Please provide references.

Response

Thank you, references have been added.

Page 9, lines 271~273

This is related to the employment environment surrounding people with schizophrenia in Korea, particularly the difficulty that people with schizophrenia experience when trying to return to their former jobs after the onset of the disorder(Chul, 2015). Most people with schizophrenia have unskilled jobs, such as labouring, manufacturing, or sales/service. Although this may be due to social bias, it also appears to be a result of the disorder becoming chronic, in addition to schizophrenia’s characteristic symptoms(Ostrow, Smith, Penney, & Shumway, 2019).

Especially in regard to the statement that patients with schizophrenia have unskilled jobs: is this the case also before the disease onset? Did this study collect data about employment type before disease onset?

Response

The post-occurrence change is unknown because we did not collect occupation and employment type before onset. However, according to a previous study (Ostrow, Smith, Penney, & Shumway (2019)), many such individuals with psychiatric disabilities often end up in unskilled jobs with little opportunity for advancement after onset.

5) Page 9, lines 274-275: “thus, it can be seen that rehabilitation had a more direct impact on job retention.” How can this be seen? Please clarify.

Response

This study takes a logical leap forward to consider rehabilitation as more important than onset age because the relationship between job retention and age of onset was not found to be significant. Thank you for the comments, which helped me correct the former explanation.

Page 9, lines 280~283

Most of our participants’ age of onset was in their 20s and 67% had a duration of illness of 10 years or over; thus, it can be seen that rehabilitation had a more direct impact on job retention than the age of onset did.

This implies that there was no significant difference in the age of onset between the occupational maintenance group and the non-maintenance group among the study subjects. This is because the degree of social skill acquisition and education were related to vocational proficiency. This led to the similar results seen in the two groups representing the occupation type.

6) Page 10, lines 304-308: Probably because these patients had less symptoms? Please add a sentence or two on that.

Response

Thank you. It has been added as per your suggestion.

Page 10, lines 311~318

In terms of vocational characteristics, the groups showed significant differences in terms of types of employment, income, and work hours per day. A longer job retention period was associated with a higher rate of independent employment and a lower rate of sheltered employment, which aligns with previous study findings that higher income and longer work hours were associated with longer job retention (Bond et al., 2008; Park & Park, 2019).

Marwaha and Johnson (2004) report that as psychotic symptoms relapsed and intensified, occupational skills became significantly reduced, and led to job discontinuation. Also, from this study, it is eviednt that the psychotic symptoms of the subjects who remained in the job for more than 6 months were better than those who did not.

Reviewer 2 Report

Please see in the letter

Author Response

Abstract:
ï‚· The use of the “theory of mind” is not clear.
ï‚· Please clarify- social cognition includes the theory of mind?

Response Thank you for the suggestion. The “Theory of Mind” defines the ability of the mind to represent the mental states of others’ intentions (Green et al. 2008).

Theory of Mind is a one of domains of social cognition. The domains of social cognition include empathy, Theory of Mind, and attribution style. In this study, the three domains of social cognition were measured to confirm social cognition. From the results of the study, it was found that the higher the Theory of Mind score, that is, the ability to grasp the intentions of others, the longer the job retention period(Lysaker, Dimaggio, & Brüne, 2014).

ï‚· The suggestion is to improve the theory of mind? Please clarify this point

Response It has been modified as follows.

Page 1, line 21

The higher Theory of Mind means the higher the ability to grasp the intentions of others. The higher Theory of Mind and the lesser attribution style and the lesser psychotic symptoms were related with the longer is the period of job retention.

Introduction:
ï‚· You begin the Intro with the term “Social cognition”. Please provide a
definition for this term.

Response Thank you. It has been added.

Page 1, line 29

Social cognition and job retention have been found to be affected by the symptoms and functional restrictions of schizophrenia(Silberstein & Harvey, 2019). Social cognition refers to the mental operations underlying social interactions, which include the human ability to perceive the intentions and dispositions of others (Brothers, 1990).

Page 2, line 74

social cognition (i.e. The domains of social cognition include empathy, Theory of Mind, and attribution style. empathy is defined as the ability to understand as well as sympathise with the experience and emotion of others (Van Donkersgoed et al., 2015). Theory of Mind refers to the ability to represent the mental states of others’ intentions. Attributional style is assigning causality to positive and negative events, which may emphasize ambiguous situations (Green et al. 2008)).

Methods:
ï‚· “Cronbach’s alpha internal consistency for the study sample was .88”- to
which scale?

Response Thank you. For pointing this out It was identified, but deleted as a sentence irrelevant to the context.

Page 3, line 118

Cronbach’s alpha internal consistency for the study sample was .88.

ï‚· Brief Psychiatric Rating Scale (BPRS)- how did you calculate the index?
Please clarify how did you calculate the indexes for all the scales you used.

Response The answers to questions 1 to 18 (not evaluated, 0 to 4 points) are converted into 0 → 1 point, 1 → 2 points, 2 → 3 points, 3 → 4 points, 4 → 5 points, and the total score is converted. The lower the score, the milder the psychotic symptoms. The participants were 30 in all comprising 22 mental health professionals, 1 nurse, and 7 social workers. Their experience in the field of mental health averaged 5 years and 6 months; they were persons who had more than 6 months’ work experience as case managers of the subjects in the institution. To ensure that data collectors did not find it difficult to use research tools, guidelines were prepared and distributed after training the collectors.

ï‚· Can you provide more information regarding the interviews and observation
you performed regarding the mental health professional data?

Response First, this study was conducted after IRB deliberation, and data collectors (case managers) are also classified as personal information protection targets. Hence, we are unable to present the photos and raw data related to the requested training and interview.

However, within the allowed range, codes the data collector's information in an anonymous form has been attached.

ID

gender

age

month of work

occupation

current service

patient number in charge

1

2

25

18

2

6

50

2

2

46

234

1

90

48

3

2

27

46

2

6

50

4

2

36

105

2

81

13

5

2

38

173

2

72

10

6

2

33

106

2

13

10

7

1

27

12

2

12

10

8

2

39

199

1

51

21

9

2

36

120

2

7

10

10

1

43

48

5

48

16

11

2

32

87

2

30

13

12

2

32

81

2

81

15

13

2

29

1

5

1

10

14

2

26

1

5

1

10

15

2

32

106

2

33

16

16

2

25

6

2

6

16

17

2

27

29

5

20

16

18

2

26

32

5

20

15

19

2

38

4

4

4

40

20

2

27

14

2

14

8

21

2

31

42

2

42

12

22

2

37

133

1

34

15

23

1

29

18

2

6

50

24

2

39

114

1

48

45

25

2

26

8

3

8

48

26

2

28

6

2

6

49

27

1

28

34

5

34

48

28

1

29

34

5

34

48

29

2

31

73

2

73

61

30

2

39

120

1

57

35

Variables

Abbreviation

Contents

Variable value

Number

ID

subjects number

1~30

Gender

gender

gender

1: male, 2: female

Age

age

age

actual number

Month of mental health service work experience

month of work

month of mental health service work experience

actual number

Type of occupation

occupation

Type of occupation

1:mental health nurse, 2: mental health social worker, 3: mental health clinical psychologist, 4: registered nurse, 5: social worker, 6: clinical psychologist

The period of current service

current service

The period of current service

actual number

Number of patients in charge

patient number in charge

Number of patients in charge

actual number

ï‚· In the Measures please provide info. Regarding the demographic
characteristics.

Response Thank you for pointing this out. It has been added.

Page 3, line 115

The demographic characteristics of this study were identified by 6 questions asking about the gender, age, onset of age, number of psychotic episodes, marital status, and level of education. Vocational characteristics were captured suing four items: previous employment, occupation and employment type, income, and length of work day.

The clinical characteristics measured in this study comprised psychotic symptoms,

Discussion:
ï‚· Please provide 2 more limitations of the study.

Response Thank you for pointing this out. I have added it as follows.

Page 11, line 369

A limitation in this study relates to our use of a sample of Korean people living in specific areas; thus, the study results cannot be generalised to people living in different areas in Korea or other countries.

In addition, it is necessary to identify physical illnesses, such as metabolic syndrome and family support, and suggest follow-up studies to determine the degree of impact of job retention on the quality of life of schizophrenic patients.

References:
ï‚· Add the year- Laurie K, Brassington L, Hodgson A, Mitchell G, Hynie A,
Kilfedder C, et al. Evaluation of a Job Retention and Vocational
Rehabilitation Pilot in Fife.

Response Thank you. Publication year has been added.

Page 12, line 398

9. Laurie K, Brassington L, Hodgson A, Mitchell G, Hynie A, Kilfedder C, et al. Evaluation of a Job Retention and Vocational Rehabilitation Pilot in Fife. 2008.

Reviewer 3 Report

The manuscript titled “Effects of psychotic symptoms and social cognition on job retention in patients with schizophrenia in Korea” presented an original research article on the relationship between psychotic symptoms, social cognition and job retention among people with schizophrenia in Korea. One hundred and fifty-eight people diagnosed with schizophrenia were divided into two groups: the first group with 75 participants had a job retention period of less than 6 months, the second group with 83 people had a job retention period of 6 months or more. All the participants were administered tools assessing psychotic symptoms, global and interpersonal relationship functioning, and three components of social cognition: empathy, theory of mind, and attribution style. A logistic regression model was used to analyze data. Results showed that psychotic symptoms, theory of mind and attribution style played a significant role in job retention. Authors discussed their results in light of previous literature as well as practical implications of their research.

I carefully read the manuscript, and I think it may be of interest for the readers of IJERPH. I think there are some minor issues to be addressed before publication. Below there are my comments and suggestions.

The research question of this study is well defined and represent an advancement in the field of epidemiology of mental illnesses, and particularly in the investigation of clinical and psychological factors which could play a significant role in job retention in people with schizophrenia. The paper is well written and the English-language used is clear. Methods and Results sections are clear and their contents are well-organized. The results are appropriately reported and interpreted; statistics used are pertinent with the aims of the study. Tables are understandable and well laid out. Discussion follows a commendable logic which elaborates the rationale explained in the Introduction.

Introduction section

I found Introduction section very clear and informative about previous literature on job retention and professional outcomes in people with schizophrenia. I have just one suggestion: Page line 70, Authors wrote “(i.e. empathy—which is defined as the ability to understand as well as sympathise with the experience and emotion of others [16]—theory of mind, and attribution style)”. Please add a sentence with a brief definition and a reference for theory of mind and attribution style, as well as you did with respect to empathy.

Material and Methods

Page 2 line 85, Author wrote “people with language disorders, substance abuse problems, intellectual disabilities, or who had been diagnosed with either neurocognitive disorders or epilepsy were excluded”. I understand the need to work with data coming from a homogenous and “pure” sample, but we have to be aware that comorbidities are very frequent in schizophrenia. It would be interesting to know what were the percentages of people suffering with schizophrenia AND other conditions which were excluded from the final sample. Moreover, as a consequence of these exclusion criteria, results can be generalized only to people diagnosed exclusively with schizophrenia, and this could be posed as a limitation.

Page 2 line 89, Authors wrote “Couture et al. [19] conducted statistical analyses on participants using a significance level (α) of .05, intermediate effect size of .15, and statistical verification of .80. Considering the dropout rate, 157 participants were needed, and allowing for non-response from 30%, a sample size of 210 was chosen for statistical analyses in this study”. I really appreciate that Authors conducted a power analysis in order to determine the sample size needed to test their hypotheses minimizing type II error, but from these lines it is unclear whether the power analysis was related to their study or to the study of Couture et al. Moreover, it should be specified a) what is the specific effect size considered (e.g., Cohen’s d, partial eta-square, etc.) and b) what kind of analysis was chosen from the options provided by the software.

Page 6 Table 1, section “Level of education (years)”: “≤16” should be “≥16”

Page 6 line 192, Authors wrote “The differences between the two groups in terms of psychotic symptoms (r = -0.33, p < .001), global functioning (r = 0.37, p < .001), and interpersonal functioning (r = 0.22, p = .007) were significant”. It is unclear why Authors reported differences between groups using a “r” statistic, while in Table 2 Student’s t is reported. Please, could Authors provide an explanation for this?

The same question as above goes for lines 199-200 and for lines 210-212, which refer to results provided in Table 3 and 4, respectively.

Page 7 line 222, Authors wrote “The explanatory power of the dependent variables in the regression model”. It should be “independent variables” instead of “dependent variables”.

Page 7 lines 232-239: Please, reframe the paragraph when you explain the meaning of an odd ratio, since I found it unclear and difficult to read. For example, at one line you wrote “When the score for hostility under attribution style increased by 1, there was a 0.948 times decrease in job retention of 6 months or more.”, and just below you wrote another interpretation of the same coefficient (0.948), this time related to another variable: “Inversely, for the direction of decrease in scores for hostility under attribution style, Exp(B) was calculated as 2.506 (1/0.399), and for psychotic symptoms Exp(B) was calculated as 1.054 (1/0.948).”

Discussion section

Page 10 line 331, Authors wrote “The attribution style score for the group with job retention of 6 months or more was close to that of non-disabled people in previous studies, and was lower than that of people with supported employment”. What you write is true, but it is true that also the attribution style score of the less than 6 months job retention group is a) similar to that of non-disabled people in previous studies, and b) lower than scores obtained by people with supported employment. How can you explain this result?

Conclusion section

Page 11 line 351: I would delete the sentence “Among people with schizophrenia, job retention was found to be impacted by theory of mind, use of hostile attribution style, and psychotic symptoms.” since it is redundant with respect to the sentences reported above.

Minor issues

Please carefully check the manuscript for typos and minor grammar issues.

Author Response

Introduction section

I found Introduction section very clear and informative about previous literature on job retention and professional outcomes in people with schizophrenia. I have just one suggestion: Page line 70, Authors wrote “(i.e. empathy—which is defined as the ability to understand as well as sympathise with the experience and emotion of others—theory of mind, and attribution style)”. Please add a sentence with a brief definition and a reference for theory of mind and attribution style, as well as you did with respect to empathy.

Response Thank you for your careful review. As per your suggestion, an explanation has been added.

Page 2, line 74

social cognition (i.e. The domains of social cognition include empathy, Theory of Mind, and attribution style. Empathy is defined as the ability to understand as well as sympathise with the experience and emotion of others (Van Donkersgoed et al., 2015). Theory of Mind refers to the ability to represent the mental states of others’ intentions. Attributional style is assigning causality to positive and negative events that may emphasize ambiguous situations (Green et al. 2008)).

Material and Methods

Page 2 line 85, Author wrote “people with language disorders, substance abuse problems, intellectual disabilities, or who had been diagnosed with either neurocognitive disorders or epilepsy were excluded”. I understand the need to work with data coming from a homogenous and “pure” sample, but we have to be aware that comorbidities are very frequent in schizophrenia. It would be interesting to know what were the percentages of people suffering with schizophrenia AND other conditions which were excluded from the final sample. Moreover, as a consequence of these exclusion criteria, results can be generalized only to people diagnosed exclusively with schizophrenia, and this could be posed as a limitation.

Response The exclusion criteria for the study subjects were those who were unable to perform the minimum function required to complete the questionnaire in this study, and there may have been comorbid conditions. It is difficult to say that this study is only for patients diagnosed with “pure” schizophrenia and that it excludes all comorbid diseases. This part has been modified as follows.

Page 2 line 91

people who were unable to read and answer the questionnaire, had language disorders, substance abuse problems, intellectual disabilities, or who had been diagnosed with either neurocognitive disorders or epilepsy were excluded.

Page 2 line 89, Authors wrote “Couture et al. (Berger et al.) conducted statistical analyses on participants using a significance level (α) of .05, intermediate effect size of .15, and statistical verification of .80. Considering the dropout rate, 157 participants were needed, and allowing for non-response from 30%, a sample size of 210 was chosen for statistical analyses in this study”. I really appreciate that Authors conducted a power analysis in order to determine the sample size needed to test their hypotheses minimizing type II error, but from these lines it is unclear whether the power analysis was related to their study or to the study of Couture et al. Moreover, it should be specified a) what is the specific effect size considered (e.g., Cohen’s d, partial eta-square, etc.) and b) what kind of analysis was chosen from the options provided by the software.

Response The below has been corrected based on the insightful comments of the reviewers.

Page 3, line 95

Statistical power analyses of the number of participants was done by using the G*Power 3.1 program (Faul et al., 2007). Couture et al. (2006) conducted statistical analyses on 139 participants using a significance level (α) of .05, intermediate effect size of .15, and statistical verification of .80. Considering the dropout rate, 157 participants were needed, and allowing for non-response from 30%, a sample size of 210 was chosen for statistical analyses in this study.

Using the G * Power 3.1 program, the significance level (α) of .05 and the statistical power of .80 were input to calculate the number of subjects required for logistic regression analysis, which was found to be 153 people. In a previous study (Bae et. Al. 2016) on Korean mentally ill patients, 30% of respondents were poor or non-responsive. In this study, considering the 30% dropout rate of the previous study (Bae et. Al. 2016), questionnaires were distributed to 210 people, and 165 surveys were collected.

Page 5 Table 1, section “Level of education (years)”: “≤16” should be “≥16”

Response Thank you. Table 1. Corrected

Table 1.

Variables

Categories

Total(n)

Less than 6 months

More than 6 months

χ²

p

n (%)

M ± SD

n (%)

M ± SD

Level of education (years)

9–11

17

11 (64.7)

6 (35.3)

4.067

0.254

12–15

101

49 (48.5)

52 (51.5)

16

40

15 (37.5)

25 (62.5)

Page 6 line 192, Authors wrote “The differences between the two groups in terms of psychotic symptoms (r = -0.33, p < .001), global functioning (r = 0.37, p < .001), and interpersonal functioning (r = 0.22, p = .007) were significant”. It is unclear why Authors reported differences between groups using a “r” statistic, while in Table 2 Student’s t is reported. Please, could Authors provide an explanation for this?

Response The addition of the correlation coefficient "r" to the text was added for the readers’ ease of understanding. However, considering the reviewer's comment and confirming a number of papers including logistic regression analysis (Kiejna et al., 2015), I realized it was inappropriate to present the correlation coefficient "r" and thus corrected it.

Page 5~6

3.2. Clinical Characteristics

The differences between the two groups in terms of psychotic symptoms (p < .001), global functioning (p < .001), and interpersonal functioning (p = .007) were significant.

The same question as above goes for lines 199-200 and for lines 210-212, which refer to results provided in Table 3 and 4, respectively.

Response I understand the reviewer’s point and realized it was inappropriate to present the correlation coefficient "r" and made the correction accordingly.

Page 6~7

3.3. Vocational Characteristics

This group also had a higher income and longer work hours in comparison to the group with a job retention period of less than 6 months. Differences between the two groups in terms of employment type (p = .011), income (p = .003), and work hours (p = .009) were significant.

3.4. Social Cognition

In terms of social cognition, differences between the two groups in terms of theory of mind (p < .001), blame (p = .001), and hostility (p = .005) were significant.

Page 7 line 222, Authors wrote “The explanatory power of the dependent variables in the regression model”. It should be “independent variables” instead of “dependent variables”.

Response Thank you. It has been modified as an independent variable.

Page 8, line 232

The explanatory power of the independent variables in the regression model was 52.7%, and the classification accuracy of the two groups in the regression model was 82.9%.

Page 7 lines 232-239: Please, reframe the paragraph when you explain the meaning of an odd ratio, since I found it unclear and difficult to read. For example, at one line you wrote “When the score for hostility under attribution style increased by 1, there was a 0.948 times decrease in job retention of 6 months or more.”, and just below you wrote another interpretation of the same coefficient (0.948), this time related to another variable: “Inversely, for the direction of decrease in scores for hostility under attribution style, Exp(B) was calculated as 2.506 (1/0.399), and for psychotic symptoms Exp(B) was calculated as 1.054 (1/0.948).”

Response  Thank you for pointing this out. In the sentence below, ‘Inversely,’ which causes confusion has been deleted to correct the meaning.

Page 8, lines 242-245

showing that when the score for theory of mind under social cognition increased by 1, there was a 1.468 times increase in job retention of 6 months or more. When the score for Psychotic symptoms increased by 1, and hostility under attribution style increased by 1, there was a 0.948 times and a 0.399 times decrease in job retention of 6 months or more.

Discussion section

Page 10 line 331, Authors wrote “The attribution style score for the group with job retention of 6 months or more was close to that of non-disabled people in previous studies, and was lower than that of people with supported employment”. What you write is true, but it is true that also the attribution style score of the less than 6 months job retention group is a) similar to that of non-disabled people in previous studies, and b) lower than scores obtained by people with supported employment. How can you explain this result?

Response Thank you. This has been corrected.

Page 10,  line 341

In this study, the job retention group over 6 months was found to have a lower attribution style score than the group with a job retention period of less than 6 months. However, both groups had scores similar to that of non-disabled people in previous studies and lower than that of people with supported employment.

Conclusion section

Page 11 line 351: I would delete the sentence “Among people with schizophrenia, job retention was found to be impacted by theory of mind, use of hostile attribution style, and psychotic symptoms.” since it is redundant with respect to the sentences reported above.

Response Thank you for the suggestion. It has been deleted.

Page 11 line 362

Among people with schizophrenia, job retention was found to be impacted by theory of mind, use of hostile attribution style, and psychotic symptoms.

This study has importance in that we showed that social cognition should be included in job retention interventions in relation to the influence of types of employment, psychotic symptoms, and function. Specifically, for stable job retention among people with schizophrenia, an intervention program is needed to improve social cognitive abilities, particularly in relation to addressing psychotic symptoms and attribution bias under social cognition, and improving flexibility of thought and empathy in interpersonal relationships.

Minor issues

Please carefully check the manuscript for typos and minor grammar issues.

Response Thank you again for the meticulous review. The manuscript has been carefully checked and corrected again.
